# Personalised high tibial osteotomy has mechanical safety equivalent to generic device in a case–control in silico clinical trial

Alisdair R. MacLeod [1], Nicholas Peckham [2], Gil Serrancolí [3], Ines Rombach[2], Patrick Hourigan[4], Vipul I. Mandalia[4], Andrew D. Toms[4], Benjamin J. Fregly[5] & Harinderjit S. Gill [1,6 ✉]

## Abstract

**Background** Despite favourable outcomes relatively few surgeons offer high tibial osteotomy (HTO) as a treatment option for early knee osteoarthritis, mainly due to the difficulty of achieving planned correction and reported soft tissue irritation around the plate used to stablise the osteotomy. To compare the mechanical safety of a new personalised 3D printed high tibial osteotomy (HTO) device, created to overcome these issues, with an existing generic device, a case-control in silico virtual clinical trial was conducted.

**Methods** Twenty-eight knee osteoarthritis patients underwent computed tomography (CT) scanning to create a virtual cohort; the cohort was duplicated to form two arms, Generic and Personalised, on which virtual HTO was performed. Finite element analysis was performed to calculate the stresses in the plates arising from simulated physiological activities at three healing stages. The odds ratio indicative of the relative risk of fatigue failure of the HTO plates between the personalised and generic arms was obtained from a multi-level logistic model.

**Results** Here we show, at 12 weeks post-surgery, the odds ratio indicative of the relative risk of fatigue failure was 0.14 (95%CI 0.01 to 2.73, $p = 0.20$).

**Conclusions** This novel (to the best of our knowledge) in silico trial, comparing the mechanical safety of a new personalised 3D printed high tibial osteotomy device with an existing generic device, shows that there is no increased risk of failure for the new personalised design compared to the existing generic commonly used device. Personalised high tibial osteotomy can overcome the main technical barriers for this type of surgery, our findings support the case for using this technology for treating early knee osteoarthritis.

## Plain Language Summary

Surgical treatment to realign the knee, called a high tibial osteotomy, is effective at relieving symptoms of knee osteoarthritis but the operation is difficult. A new personalised treatment with simpler surgery has been designed. The aim of this study was to investigate the safety of the new personalised treatment compared to the standard treatment. For the first time, a detailed computer simulation clinical trial was performed, using imaging data from 28 real patients. The computer simulation compared the risk of the implant failure between the personalised and standard treatments. The personalised treatment did not have a higher risk of implant failure than standard treatment. This supports further clinical studies looking at the benefits of personalised over standard realignment surgery. The personalised treatment has the potential to allow much more widespread use of realignment surgery to treat early knee osteoarthritis.

[1] Department of Mechanical Engineering, University of Bath, Bath, UK. [2] Oxford Clinical Trials Research Unit, NDORMS, University of Oxford, Oxford, UK. [3] Department of Mechanical Engineering, Polytechnic University of Catalonia, Barcelona, Catalunya, Spain. [4] Royal Devon and Exeter NHS Foundation Trust, Exeter, UK. [5] Department of Mechanical Engineering, Rice University, Houston, TX, USA. [6] Centre for Therapeutic Innovation, University of Bath, Bath, UK. ✉email: r.gill@bath.ac.uk

The lifetime risk of knee osteoarthritis (OA) is estimated to be as high as 45%[1] and is becoming more common[2]. Though the demand for knee replacement is predicted to double by 2030[3], it is only suitable for end-stage disease[4,5]. Knee replacement is non-reversible as it involves removing the natural joint structures and replacing these with metal and plastic; this has an impact on function. Up to 30% of knee replacement patients report being unsatisfied with their surgery[6], with younger patients having higher rates of revision and greater levels of dissatisfaction[7]. Alternatives to knee replacement for osteoarthritis treatment are urgently needed to reduce individual suffering as well as reducing the financial and societal burden of knee OA. High tibial osteotomy (HTO) is an established and effective[8] knee preserving treatment for early-stage OA and has even been used successfully for more advanced OA[9]. The HTO procedure involves creation of an opening or closing wedge osteotomy in the proximal tibial to change the varus alignment, thereby altering the mechanical axis of the leg and reducing the load in the painful compartment[10]. Alignment is usually measured in the frontal plane using the hip–knee–ankle (HKA) angle. The osteotomy is commonly stabilised using an osteosynthesis plate, though hemicallotasis with an external fixator is also sometimes used to introduce the alignment change gradually.

Opening wedge HTO procedures are more popular[11] due to the simpler surgical approach, lower risk of peroneal nerve damage[12,13] and better post-operative flexion scores[13]. Medial opening wedge HTO has been reported to have survival rates between 82%[9] and 94%[14] at 10 years. Numerous studies have shown that the long-term outcomes are related to the accuracy of the surgical correction achieved relative to the planned correction[12,15]. Van Den Bempt et al.[15] undertook a systematic review covering the topic of accuracy and concluded that "*accuracy of coronal alignment corrections using conventional HTO falls short*". The mean value of accuracy (=mean value of the error between the desired correction angle and the achieved correction angle) in coronal plane alignment correction was approximately 6° (range 4°–8°). The difficulty in achieving the planned correction is a concern for surgeons given that outcome is dependent upon accuracy of correction and is cited as a key factor for why some knee surgeons do not offer HTO to their patients.

Patients with delayed consolidation of the osteotomy have been reported to have a statistically significant greater risk of complications[16]. In addition, studies report better outcomes, including a lower complication rate, when using angularly stable plates compared to smaller spacer plates[17–20]. Pain and discomfort due to the plate is common and in some centres, it is routine to remove the plate after a follow-up period of 12 months[21]. However, a significant proportion of individuals (7.2–23%) require earlier plate removal due to pain from soft-tissue irritation[20,22]. It is important to ensure that all plates have sufficient flexibility to promote bone healing[23].

Patient-specific HTO procedures directly address the limitations of generic HTO procedures, overcoming the issues of accuracy, excessive stiffness and soft-tissue irritation. Patient-specific surgical guides have been demonstrated to increase accuracy of correction. Munier et al.[24] reported a difference of <2° between the planned and achieved HKA correction in a cohort of ten patients using patient-specific guides. A recent cadaveric study[25] has demonstrated that combining 3D printed patient-specific plates with metal 3D printed patient-specific surgical guides further improves accuracy, reducing the difference between planned and achieved HKA to ~0.5°. Patient-specific plates have a unique advantage in being able to optimise locking screw orientations for individual bone geometries, whereas achieving fixation with generic plates is compromised by having generic fixed locking screw orientations. Digital 3D planning based on individual patient tibial geometry allows screw lengths and orientations to be determined pre-operatively and embodied in the surgical guide. Mathews et al.[25] demonstrated that the operative procedure was greatly simplified and operative times of 30 min readily achievable.

Plates designed based on individual tibia geometry also offer the capability to exactly match the surface of the patient's proximal tibia, thus minimising the likelihood of soft-tissue irritation. The advent of additive manufacturing in metal makes personalised surgical guides and plates an economically viable option. As each plate is unique, there is variability in plate shape due to individual anatomical differences. This situation could potentially result in a higher variation in plate stress compared to a generic plate. On the other hand, there is also the potential to have more consistency—adaptive sizing specific to the patient's size and weight has the potential to improve consistency of outcomes. To our knowledge, there has been no biomechanical comparison of a patient-specific HTO plate against a generic plate for a cohort of patients. There have been cadaver trials to evaluate procedural aspects of patient-specific HTO implants[26], but no study has evaluated performance during clinically relevant physiological activities. Patient-specific HTO procedures with personalised HTO plates and surgical guides overcome the principal limitations of HTO surgery and can potentially make this joint preserving surgery more widely available, however, it is important to establish that the risk of mechanical failure is not greater than that for generic plates.

Surgical clinical trials use comparison between a new treatment arm and an established (control) treatment arm, where avoiding bias between treatment arms is always a concern. Computational modelling has the potential to introduce a new paradigm—the ability to simulate multiple surgeries on virtual copies of the same individual to compare on a paired basis the mechanical outcomes between new and established interventions. For case–control study design, an in silico clinical trial enables each virtual participant to be their own control.

The aim of the current study was to compare subject-specific high tibial osteotomy plates and the most commonly used established generic plates in terms of mechanical function, and hence risk of failure, by conducting a case-control in silico clinical trial with a clinically relevant knee osteoarthritis cohort. As far as possible the conventions established for physical clinical trials were followed, with the aim of transparent reporting. The trial was registered at ClinicalTrials.gov (clinicaltrials.gov/ct2/show/NCT03419598).

The principal outcome measure was the peak mechanical stress present in the implanted plates during physiological loading as determined by finite element analysis (FEA). Stability of the osteotomy was assessed by calculating the interfragmentary motion. The key finding is that there is no increased risk of failure for the new personalised design compared to the existing generic commonly used device.

## Methods

**Study design**. This was designed as a case–control in silico clinical trial.

**Setting**. Patient data (computed tomography [CT] scans and demographics) were obtained from patients with radiologically confirmed knee osteoarthritis presenting at a specialist orthopaedic centre (Princess Elizabeth Orthopaedic Centre, Royal Devon and Exeter NHS Foundation Trust, Exeter, UK). Patient data collation took place from January 2017 to March 2018; data were anonymised and transferred to University of Bath for segmentation, geometric model creation, virtual surgery, finite element model creation, calculation and application of physiological loads and finite element model solution. Finite element models were solved using the Balena High Performance Computing

Service at the University of Bath. The simulation time points were 2, 4 and 12 weeks post surgery.

**Participants**. Ethical approval was obtained to anonymously use CT scans of 30 patients with moderate to severe knee arthritis (REC reference: 17/HRA/0033, RD&E NHS, UK). Informed consent was not needed as this study was granted ethics approval to have anonymous re-use of existing data. The inclusion criteria were:

- Appropriate existing CT data of lower limb.
- Male or Female, aged 18 years or above.
- Diagnosed with moderate to severe OA of the knee.

Exclusion criteria were:

- Abnormal anatomy of tibia or presence of pathology other than OA, e.g. bone tumour.
- Previous knee or osteotomy surgery.
- Presence of metal-work around the knee.

Due to poor CT scan quality, two patients were disqualified, leaving a cohort of 28 patients. Patients were 50–87 years old (mean: 68), 54% female, 68.8–121.4 kg (mean: 90.1 kg), 147–190 cm tall (mean: 169 cm) and had no history of knee surgery (Supplementary Table S1).

**Power study**. A power analysis was performed using the experimentally measured variation in stiffness and strength for standard-sized TomoFix HTO plates[27]; the TomoFix HTO plate (DePuy Synthes, IN, USA) is a widely implanted HTO device and was used as the generic HTO device in this study. A previous experimental study measuring stiffness of TomoFix bone-plate constructs found the mean and standard deviation to be 1950 N/mm and 577 N/mm respectively. Based on these values and the method of Altman[28], 25 patients per arm would be needed to give the study 80% power for a detectable difference of 20% in stiffness. For a virtual clinical trial, since the same patient can particulate in both arms, only 25 patients total were required.

**Correction assessment and intervention**. The CT data were used to generate the 3D geometry of each patient's proximal tibia (ScanIP M-2017.06, Synopsys Inc., CA, USA). Five key landmarks of interest were identified on the CT scan (Supplementary Fig. S1), and the osteotomy correction angle required was calculated (Matlab R2017b, MathWorks, MA, USA) such that the altered mechanical axis passed through a point 62.5% of the distance from medial to lateral tibial plateau[18]. The calculated correction angle for each patient is given in Supplementary Table S1. Virtual HTO surgery was performed on each patient to alter the mechanical axis of the knee by creating an opening wedge osteotomy (ANSYS SpaceClaim R18.2, ANSYS Inc., PA, USA) with the guidance of an orthopaedic surgeon specialising in knee surgery. The medial opening wedge osteotomy was placed at an angle of 15° to the tibial plateau; the lateral bone hinge was located at least 10 mm below the joint line.

After the virtual surgeries were performed, each virtual patient was duplicated. One copy had the osteotomy stabilised using the Generic plate and the other had the osteotomy stabilised using the Personalised plate, thus forming the two arms of the trial (the geometry of both plates are shown in Supplementary Fig. S2):

| Arm A | Generic=HTO stabilised with a generic osteotomy plate (Tomofix, Depuy Synthes) |
| Arm B | Personalised =HTO stabilised with a patient-specific osteotomy plate |

For the Generic arm, the TomoFix plate geometry was generated from a micro-CT (H 225 ST, Nikon Metrology inc.,

USA) scan of a physical TomoFix medial high tibia plate (standard size, model number 440.834) using image processing software (ScanIP M-2017.06). For the Personalised arm, the patient-specific implant geometries were generated using specialised planning software (Renishaw plc, Wotton-under-Edge, Gloucestershire, UK), taking into account the surface of the tibia and the degree of correction for each patient. All simulated knees were virtually implanted with both implant types, thereby generating 56 intervention cases as finite element models (ANSYS 18.2, ANSYS Inc.).

**Finite element models: material properties**. Finite element models were created and FEA performed based on a validated methodology and model[29]. The modelling parameters from the validated model were used, including the method of representing the screws and plate as well as contact interactions between the components. In the current study, patient-specific material properties were applied from each patient's CT data (BoneMat 3.2, Istituto Ortopedico Rizzoli, 2015) using heterogeneous linear elastic properties defined by the following relationship ($HU$ = Hounsfield Unit, $\rho_{CT}$ = CT based density, $\rho_{Ash}$ = ash density, $E$ = Young's Modulus):

$$\rho_{CT} = -0.00393573 + 0.000791701 \times HU \tag{1}$$

$$\rho_{Ash} = 0.079 + 0.877 \times \rho_{CT} \tag{2}$$

$$E = 14,664 \times \rho_{Ash}^{1.49} \tag{3}$$

The values are typical and within the range of values found in the literature[30]. Approximately 240 values for $E$ were used in the proximal bone fragment and 450 in the distal fragment. Average lowest $E$ values were 350 MPa and highest were 23 GPa distally and 13 GPa proximally.

At the plate-screw interface, normal contact stiffness was set to 0.002, determined on the basis of experimental testing. A standard coulomb friction coefficient of 0.8 for the tangential behaviour and an Augmented Lagrange contact formulation were used. All other contacting surfaces (screw-bone, bone–bone) were assumed to be bonded with ANSYS default contact settings.

The progression of bone healing was represented by increasing the Young's modulus of the osteotomy region. Data from previous studies quantifying the extent of osteotomy gap healing at different time points[31,32] was used to inform the material characteristics selected for each phase of healing[33,34]. Supplementary Table S2 details the values selected and the phases of healing considered.

**Finite element models: meshing**. Meshing parameters were based on a previous validated study[29]. Fully integrated quadratic tetrahedral elements were used with an element size of 0.8 mm for the plate, screws, and cortical hinge regions, 1.4 mm for the rest of the bone, and 2 mm in the healing osteotomy region. The mesh was refined around the plate-screw interaction such that the average element edge length was 0.3 mm. The number of elements used was over 1 million, with ~70k per screw, 200k in the plate, and 600k in the bone. A mesh convergence study was performed and the mesh resolution was selected if there was less than a 5% change in peak Von Mises stress after doubling the number of elements. Since doubling the plate mesh resolution from 192k elements to 380k elements resulted in a 3.92% change in the peak stress, a mesh resolution of 192k elements was selected. A mesh sensitivity study was performed for the plate-screw contact stiffness properties. For the stiffness value selected, the displacement results changed by less than 3.35% for every mesh size evaluated.

**FEA physiological activities**. Muscle forces and joint reaction forces for normal physiological activities were calculated using a subject-specific musculoskeletal model (gender: male, age: 88 years, mass: 65 kg, and height: 166 cm)[35]. These forces were automatically registered to the individual patient geometries using a custom transformation and scaling script[36] (Matlab R2017b, The Mathworks, Natick, MA, USA) based on the least-squared error optimisation of five landmarks. The muscle and joint reaction forces were linearly scaled by each subject's body weight.

Three common activities were considered: (ACT1) Fast walking gait, (ACT2) Chair rise, and (ACT3) Squat. Five key instances were selected for each activity based on locations of peak tibal contact force (Supplementary Fig. S3). These instances were implemented as load steps 1 to 15 within the finite element models. The joint reactions at each of these key instances are outlined in Supplementary Table S3.

**FEA evaluated parameters**. For each patient in both arms—Generic and Personalised—of the virtual clinical trial, a variety of permutations were run (Supplementary Fig. S4). These simulations included:

Three physiological activities: (ACT1) Fast walking gait, (ACT2) Chair rise, and (ACT3) Squat;

Three screw configurations: (SC1) all screws present, (SC2) screw closest to osteotomy removed to produce a longer bridging span, and (SC3) most distal screw removed to simulate a shorter plate. The comparison of screw configurations were performed at healing stage 2 as described below.

Osteotomy gap bone healing was simulated by increasing the Young's modulus of the material in the gap for different healing stages: (HS1) immediately post-operative period (NB this healing stage was not simulated, however this was described in our ClinicalTrials.gov entry and is included here for consistency), (HS2) 2-weeks post-operatively (1 MPa), (HS3) 6-weeks post-operatively (28 MPa), and (HS4) 12-weeks post-operatively (528 MPa). Healing stages 2, 3 and 4 were compared for screw configuration 3, which was chosen based on an analysis of the effects of screw configuration and because most surgeons expressed a clinical preference for a shorter HTO plate.

A total of 4,200 load steps were defined and solved using a geometrically nonlinear analysis (Ansys 18.2, ANSYS, Inc. USA).

**Key output variables**. The principal outcome variable was the maximum Von Mises stress within the plates. In addition, the maximum Von Mises strain in the bone adjacent to the screws used to fix the plates and the inter-fragmentary movement at the osteotomy site were evaluated. A concern for all metal implants is fatigue failure, so the number of load cases for which the maximum Von Mises stress in the plate exceeded a pre-defined fatigue limit was determined as a function of healing stage for each study arm. The fatigue limit was experimentally established by performing fatigue testing on samples additively manufactured from medical grade titanium alloy (Ti-6Al-4V) using an ISO13485 certified 3D metal printing process (AM 250, Renishaw plc, Wotton-under-Edge, Gloucestershire, UK). The fatigue limit (FLIM) was found to be $200 \pm 20$ MPa, and a conservative approach was taken by choosing the lower bound, i.e. 180 MPa. For each healing stage, the number (N1) of load cases for which the maximum Von Mises stress exceeded FLIM and the number (N2) for which the maximum Von Mises was less than FLIM was recorded.

**Statistical analysis**. The effect of the three screw configurations was investigated for each arm by performing an ANOVA (Matlab 2017b) at each of the fifteen load steps applied at healing stage 2. A Bonferroni correction was applied for this analysis by dividing the alpha value of 0.05 by 15.

The primary analysis compared the Generic (control) HTO plate and the Personalised HTO plate in terms of the ratio of load steps for which the Von Mises plate stress exceeded the FLIM and those for which it did not (N1:N2). The analysis was performed using a multi-level logistic model (StataCorp. 2019. Stata Statistical Software: Release 15. College Station, TX: StataCorp LLC.) using repeated measures over healing stage (level 1) nested within patients (level 2). The clustering of observations within patients was accounted for by using a random effect for patient identifier and robust standard errors to account for hetero-scedasticity in the data. An interaction between healing stage and device was included to obtain odds ratios with corresponding 95% confidence intervals at the different time points.

Estimates for the odds ratio at healing stage 4 (HS4) were obtained using a penalised maximum likelihood logistic regression model.

The design of this in silico clinical trial enabled each subject to act as their own control. The differences (Generic—Personalised) in maximum stress, maximum strain, and inter-fragmentary motion were analysed on continuous scale multi-level regression models using the same hierarchical structure as described above.

**Reporting summary**. Further information on research design is available in the Nature Research Reporting Summary linked to this article.

## Results
The solving of the finite element models was computationally intensive requiring 500,477 core hours on the High Performance Computing cluster.

The effects of screw configuration on the Von Mises stress in each type of plate were not significant, with a large degree of overlap between the confidence intervals (Fig. 1). The average $p$ values were 0.634 (range 0.162–0.977) for the Generic plate and 0.375 (range 0.076–0.907) for the Personalised plate.

In general, the higher applied loads for activities 2 and 3 gave rise to higher values for maximum Von Mises stresses in both plates than did the loads for activity 1. For the Generic plate, the highest mean value was 465 MPa for screw configuration three (SC3) at load step 15, while for the Personalised plate, it was 547 MPa for screw configuration two (SC2) at load step 10.

The effect of healing stage was very dramatic in reducing the maximum Von Mises stresses in both sets of plates. For healing stage 2, the overall maximum value was 2,650 MPa for the Generic arm and 1,420 MPa for the Personalised arm. At healing stage 3, these maximum values decreased to 443 MPa for Generic and 545 MPa for Personalised. Finally, at healing stage 4, maximum values dropped to 243 MPa for Generic and 165 MPa for Personalised (Fig. 2). The maximum Von Mises stresses were mostly due to large negative (compressive) values of the third principal stress, with the first principal stress always being lower (Fig. 2).

For screw configuration 3, the ratios of load steps exceeding the fatigue limit for each healing stage are given in Table 1. Comparing Personalised to Generic, the odds ratio dropped from 1.80 (95%CI 0.90–3.61) for healing stage 2, to 1.25 (95%CI 0.76–2.06) for healing stage 3, and finally to 0.14 (95%CI 0.01–2.73) for healing stage 4. These values were not statistically significant.

The delta values (= Generic—Personalised) for maximum Von Mises stress also showed a large reduction as healing stage

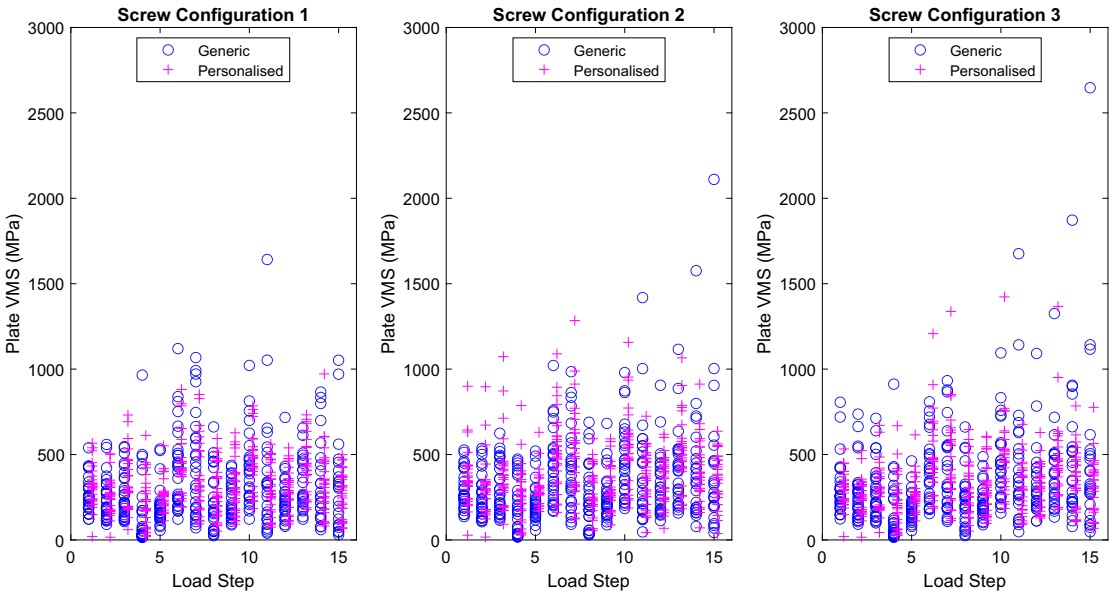

**Fig. 1 Maximum Von Mises Stress (MPa) in each type of HTO plate at healing stage 2 (HS2) for each load step for screw configurations 1, 2 and 3.** The circles are the maximum Von Mises Stress for each virtual subject implanted with the Generic plate, and the plus symbols are for the Personalised plate, at each loading step ($n = 28$ independent models for each arm, Generic and Personalised).

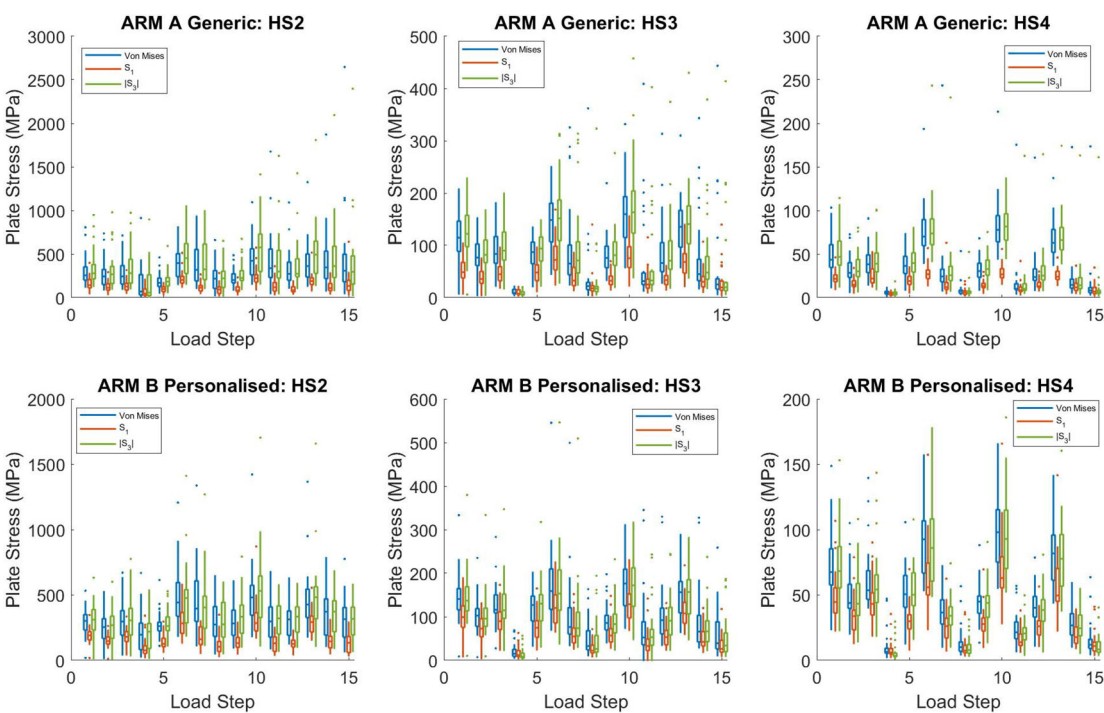

**Fig. 2 Box and whisker plots of maximum Von Mises, first principal (S1) and the modulus of the third principal stress (S3) at each load step for each arm as a function of healing stage (HS).** On each box, the central mark indicates the median, and the bottom and top edges of the box indicate the 25th and 75th percentiles, respectively, $n = 28$ independent models for each arm, Generic and Personalised. The whiskers extend to the most extreme data points not considered outliers, and the outliers are plotted individually using a dot symbol. An outlier is a value that is more than 1.5 times the interquartile range away from the bottom or top of the box.

increased (Fig. 3, Table 2). The differences in maximum Von Mises stress were not significant for healing stage 2 ($p = 0.67$) but were significant for healing stages 3 and 4 ($p < 0.001$). For healing stages 3 and 4, the mean delta value for the maximum Von Mises stress in the plate were $-17.1$ MPa (95%CI $-26.2$ to $-7.9$ MPa) and $-11.1$ MPa (95%CI $-15.3$ to $-6.8$ MPa) respectively. Expressed as a percentage of the maximum von Mises stresses in

the personalised plate for healing stages 3 and 4, these delta values represent 3.1% and 6.7% of the relevant maximum stress values and were therefore relatively small.

The differences in bone strain around the screw insertions for the two plates were small and not significant (Table 2), being in general <2 micro-strain. The differences in inter-fragmentary movement between the two devices were significant ($p < 0.001$)

**Table 1 Contingency table for number of load steps in which maximum Von Mises stress exceeded the fatigue limit (FLIM) comparing the two arms of the study, Generic and Personalised for each of the three healing stages (HS) for screw configuration 3. OR = Odds Ratio.**

| | Generic[a] | | | Personalised[a] | | | Personalised vs Generic[a] | |
|---|---|---|---|---|---|---|---|---|
| | N2: Stress< FLM | N1: Stress> FLM | Total | N2: Stress< FLM | N1: Stress> FLM | Total | OR (95% CIs) | p-values |
| HS2 | 96 (24.6%) | 295 (75.4%) | 391 | 73 (17.8%) | 337 (82.2%) | 410 | 1.80 (0.90, 3.61) | 0.10 |
| HS3 | 364 (88.8%) | 46 (11.2%) | 410 | 341 (87.0%) | 51 (13.0%) | 392 | 1.25 (0.76, 2.06) | 0.37 |
| HS4 | 413 (99.3%) | 3 (0.7%) | 416 | 419 (100%) | 0 (0%) | 419 | 0.14 (0.01, 2.73)[b] | 0.20[b] |

[a]Sata are presented for all observations, which are clustered within participants.
[b]Estimate obtained from a penalised maximum likelihood logistic regression.

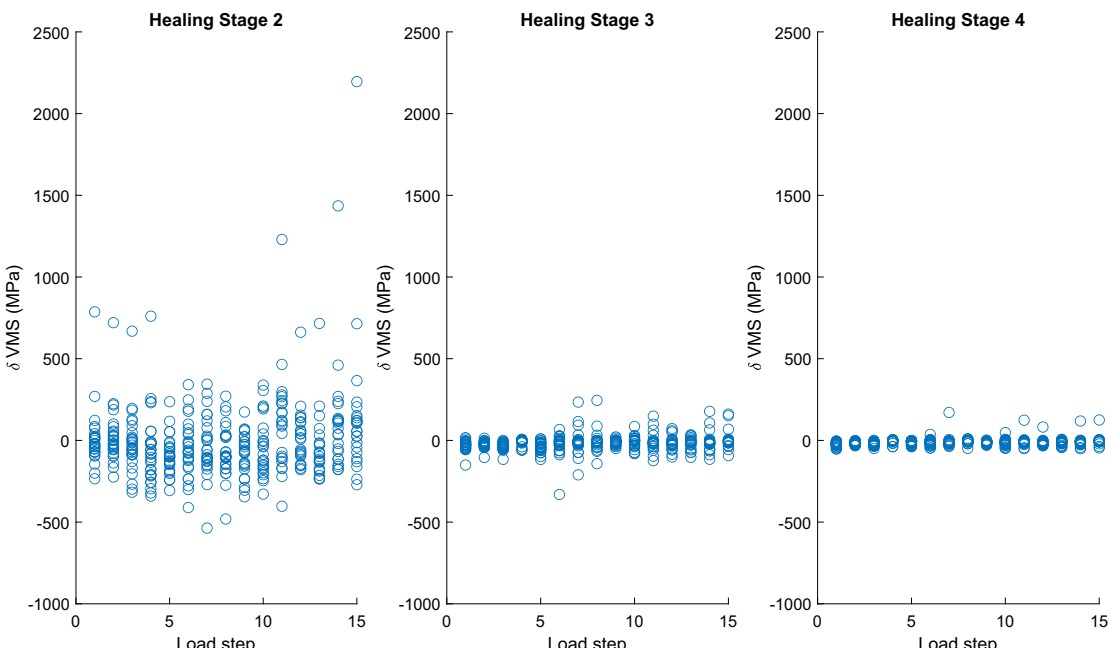

**Fig. 3 Changes in delta values for maximum Von Mises stress for each load step plotted for each of the three healing stages considered, all for screw configuration 3.** The circles are the individual differences (delta maximum Von Mises stress = Generic maximum Von Mises stress—Personalised maximum Von Mises stress) per virtual subject, $n = 28$.

only at healing stage 4 (Table 2), for which the Personalised plates exhibited more micro-motion (<0.04 mm) than did the Generic plates.

Finite element analysis is a numerical method and thus in some cases a solution cannot be achieved. A small number of load cases did not solve, giving rise to the numbers of achieved solutions in Tables 1 and 2.

## Discussion

This study set out to evaluate the mechanical function, and hence risk of failure, of a new personalised high tibial osteotomy plate compared to one of the most commonly used predicate generic devices, TomoFix, using a virtual clinical trial methodology. Stability of the osteotomy was assessed by examining the interfragmentary motion. To our knowledge, this is the first time that such a methodology has been used for an orthopaedic device. Hardware failure is a relatively rare complication using TomoFix implants[20] but is more common for other designs such as spacer plates[19]. The new personalised HTO system evaluated in this study uses patient-specific contoured plates. While pre-clinical testing is essential for such a new design to ensure safety, it is not straightforward to do for patient-specific devices. The virtual

clinical trial methodology used in this study allows the patient-specific device generation pipeline to be used in a reasonably sized ($n = 28$) cohort with knee OA, allowing the mechanical performance of a range of patient-specific devices to be compared on a paired basis with that of the TomoFix generic device. Thus, each virtual subject acts as their own matched control. This study used physiological loading representing everyday activities of fast walking, chair rise, and squatting and considered three bone healing stages. The subject-specific finite element models used for this virtual study were created based on a validated modelling pipeline[29]. As the main failure mechanism for these devices is fatigue failure, the key result is the odds ratio indicative of the relative risk of fatigue failure between the personalised and generic arms at healing stages 3 and 4. Whilst the personalised HTO plates consistently had larger stress values than the generic, the delta values were relatively small at ~3% of the maximum stress for healing stage 3 and 7% of the maximum stress for healing stage 4. In addition, at healing stage 4, the maximum stress (165 MPa) in the personalised HTO plates was well below the fatigue limit (FLIM=180 MPa). The results of this virtual clinical trial demonstrate that personalised HTO plates have no difference in the risk of failure compared to generic plates whilst being more mechanically efficient and less stiff. Personalised HTO treatment

**Table 2 Maximum von Mises stress (Stress) in the plates, maximum Von Mises strain (Strain) in the bone adjacent to the plates screws and maximum inter-fragmentary movement (IFM) for the two arms, and the differences between the arms for each of the three healing stages (HS), all cases are for screw configuration 3.**

| | | Generic[a] | | Personalised[a] | Adjusted Difference (95% CIs)[b] Generic—Personalised | |
|---|---|---|---|---|---|---|
| | n | Mean (SE) | n | Mean (SE) | | p values |
| Stress (MPa) | | | | | | |
| HS2 | 391 | 331.8 (13.55) | 410 | 345.1 (9.54) | −12.5 (−70.0, 45.0) | 0.67 |
| HS3 | 410 | 90.6 (3.56) | 392 | 108.3 (3.81) | −17.1 (−26.2, −7.9) | <0.001 |
| HS4 | 416 | 37.0 (1.64) | 419 | 48.3 (1.60) | −11.1 (−15.3, −6.8) | <0.001 |
| Strain (unitless) | | | | | | |
| HS2 | 391 | 0.015 (0.00071) | 365 | 0.017 (0.00067) | −0.0011 (−0.0030, 0.0008) | 0.27 |
| HS3 | 410 | 0.011 (0.00065) | 392 | 0.011 (0.00056) | −0.0002 (−0.0024, 0.0021) | 0.88 |
| HS4 | 416 | 0.0096 (0.00067) | 419 | 0.0090 (0.00055) | 0.0005 (−0.0020, 0.0031) | 0.67 |
| IFM (mm) | | | | | | |
| HS2 | 391 | 0.31 (0.012) | 410 | 0.33 (0.015) | −0.014 (−0.045, 0.017) | 0.37 |
| HS3 | 410 | 0.12 (0.005) | 392 | 0.12 (0.004) | −0.005 (−0.010, 0.001) | 0.10 |
| HS4 | 416 | 0.04 (0.002) | 419 | 0.06 (0.003) | −0.036 (−0.054, −0.018) | <0.001 |

[a]Data are presented for all observations, which are clustered within participants.
[b]Estimates are based on a multi-level logistic model using repeated measures over time and allowing for additional clustering within participants using robust standard errors.

with 3D printed patient-specific plates addresses the critical issues of accuracy, stiffness and geometric conformity which currently limit the use of HTO surgery with the added advantages of simpler and quicker surgery. Having shown that using personalised plates does not increase the risk of failure, it should now be possible to exploit the advantages of personalised HTO and offer joint preserving treatment for early knee OA more widely.

This virtual trial utilised a finite element modelling methodology that has now become an accepted tool for biomechanical studies, and it has been demonstrated that subject-specific finite element models created from CT data can accurately predict the mechanical behaviour of long bones[37,38]. The principal outcome measure for this study was the peak mechanical stress present in the implanted plates during physiological loading. This outcome was found to be highly influenced by healing stage. Typical time to union has been reported to range from 75 to 126 days[21,39–42]. Unfilled osteotomy gaps[41] have been associated with faster healing times, as has male gender[21]. For this reason, it was important to include the 6 week and 12 week time points as full weight-bearing may be permitted[43] without full consolidation of the healing osteotomy[31].

For both the generic and the personalised plates, the highest stresses were for healing stage 2, where the differences in plate stress between the two arms of the study were not significant (Table 2). For healing stage 3, the mean stresses decreased by a factor of 3.7 for the generic plates and 3.2 for the personalised plates. Similarly, for healing stage 4, the mean stresses decreased by a factor of 9 for the generic plates and 7.1 for the personalised plates. Nonetheless, for healing stages 3 and 4, the personalised plates still exhibited slightly higher stresses than did the generic plates, which was expected since generic devices are generally over-sized. We believe these small increases in plate stress (17 MPa and 11 MPa for HS3 and HS4, respectively) are not clinically relevant as there were no differences between the associated bone strains and only very small differences in interfragmentary micromotion (40 microns) at healing stage 4. More critical for the failure of these types of devices is whether the peak Von Mises stresses exceed the fatigue limit (FLIM = 180 MPa) for the material. This issue was examined by looking at the ratio of the number of load cases for which the fatigue limit was exceeded to the total number of load cases for each arm at each healing stage. For all healing stages, the p values associated with these odds ratios were >0.05 (Table 1) and for healing stage 4 the odds ratio was less than one.

The other outcome measures were the peak strains in the bone adjacent to the screws and the maximum interfragmentary motion at each healing stage considered. There were no significant differences in peak strain values between the two arms at any healing stage. As elevated strain levels around screws can indicate a risk of loosening[44–46], the finding of no difference between the arms indicates that there is no increased risk of screw loosening occurring with the personalised implant compared to the generic device. The mean values for the maximum interfragmentary motion were approximately 0.3 mm for both arms at healing stage 2 and ~0.1 mm for both arms at healing stage 3. The values for healing stage 3 are similar to those reported by a previous cadaver study[23], which found maximum interfragmentary motion values of ~0.15 mm under a TomoFix plate using joint loads of 1000 N (146% BW for 70 kg individual). In the current study, the maximum interfragmentary motion was only significantly different between arms for healing stage 4, where the personalised arm had a higher maximum interfragmentary motion, though the mean difference between the arms was only 36 microns (Table 2). The interfragmentary motion is an indicator of the stability of the osteotomy, and the findings show that there is no difference in stability between the personalised and generic devices. It is worth noting that that the statistical methods used accounted for correlations and clustering within the observations.

The primary outcome measure was the maximum Von Mises stress in each HTO plate calculated by each finite element analysis, which was a conservative approach as the method could potentially overestimate this value. The finite element models used included contact between the plate and the screws as well as contact between the bone and the screws. Contact modelling can be influenced by very fine geometry features of the surfaces in contact and can give rise to high compressive point stresses due to small variations in geometry. The modelling approach used was a linear elastic approach, which does not allow for plastic deformation that would occur in reality. Such plastic deformation would result in a reduction of contact stress. The compressive contact stresses is reflected in the peak Von Mises stresses. The approach of taking the peak Von Mises stress for each solved load case will therefore include the effects of high contact stresses, which will overestimate the maximum loads in the plates. Despite this issue, our findings show that by healing stage 4, the fatigue limit was exceeded by only 0.7% of the load cases for the generic device and was never exceeded for the personalised device.

Obesity has been shown to be a significant independent predictor of major complication following HTO surgery ($p = 0.001$)[42]. Consequently, use of a TomoFix plate is not recommended in obese individuals. Despite this fact, 88% of patients in a previous clinical study were overweight or obese[41]. Our study included 92.9% overweight and 78.6% obese patients and is therefore representative of the typical patient demographics.

We believe the case-control in silico trial methodology is an appropriate way to assess the mechanical function of personalised devices, particularly orthopaedic devices which have a primary load carrying function. The pipeline for producing subject specific models, however, still requires considerable user input. Robustly increasing the level of automation, whilst maintaining model fidelity, will aid in making the methodology more widely available. For the current study in order to increase transparency and avoid bias, the maximum values of calculated stress in the HTO plates were selected. As mentioned above contact conditions may give rise to high values of stress at very localised contact points. Future work should focus on robust techniques for considering these values as outliers.

It is worth mentioning the high computational requirements of this study. The patient-specific finite element models required considerable manpower resource to create them. The solutions of the finite element models required more than half a million core hours of computing time, this was only made possible by having access to a large cluster with a total number of 3544 CPU cores. The most efficient way to perform solutions was to use the 24-core Intel Skylake compute nodes, of which the cluster had 17. Utilising all 17 Skylake compute nodes equated to a computation time of approximately 52 days. The solution files generated by Ansys are very large (this study generated ~7.5 TB of solution files), and a computational framework was developed to automate the extraction of the relevant results; this added to the computational requirements of the study. A lesson learnt from performing this study is that ideally a solver with greater flexibility in generating the required result set in a compact form should be used.

This study has two key limitations. Whilst each subject-specific finite element model incorporated individual bone density, the properties used for the healing bone material (callus) at the three healing stages considered were the same for all models. Only limited amounts of information regarding these material properties are available in the literature, and future studies should look at understanding how these material properties vary in a population. There is also a paucity of detailed information regarding physiological loading. The same loading patterns, scaled by individual bodyweight, were used for the whole cohort. Investigating how to pragmatically incorporate individual loading patterns should be a priority for future studies. In the current study only a limited number of loading conditions were evaluated. Ideally, a broader range of activities and subjects would have been studied. However, the approach still provides results comparable to those of conventional clinical trial, which would have the same limitations. Furthermore, now that a paradigm for in silico clinical trials is available, a future study could explore mechanical safety and/or relative efficacy for a broader range of activities and subjects.

In this in silico trial only mechanical safety and stability were evaluated. It would be interesting to examine relative efficacy as well, which is a question that in silico clinical trials might be able to address better than conventional clinical trials. However, how to define relative efficacy quantitatively remains an open question – one that would be worth investigating in a future study.

Our findings show that the risk of mechanical fatigue failure is similar between a new personalised device for patient-specific HTO surgery and a generic widely used device. Our findings also show that the strains in the bone adjacent to the screws used to fix the HTO plates are similar for the patient specific and the generic device. The interfragmentary motion was similar for both types of device, indicating that there is no difference in the stability of the osteotomy. The results indicate that there is no increased risk of failure for the new personalised design compared to the existing generic commonly used device. With this concern addressed, the advantages of personalised HTO (better accuracy, lower stiffness, greater conformity, simpler and quicker surgery) can be exploited to make effective joint preserving treatment for early knee OA more widely available.

The use of this novel (to the best of our knowledge) in silico trial methodology provided a practical method to gain insight into the mechanical safety of personalised devices.

## Data availability

Source data for Figs. 1, 2, 3 and S3 are provided as Supplementary Data 1. The datasets generated during and/or analysed during the current study (together with muscle loading and code for transforming loads as well as the finite element model from MacLeod et al.[29]) are available in the University of Bath Research Data Archive, https://doi.org/10.15125/BATH-00926[36].

## Code availability

The custom transformation and scaling script used for transforming loads is available under CC-BY license terms from: https://doi.org/10.15125/BATH-00926.

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

## Acknowledgements

The work and lead author were funded by Arthritis Research UK, grant number 21495 and Versus Arthritis, grant 22262.

## Author contributions

Conception H.S.G., A.R.M., A.T. and P.H., acquisition A.R.M., P.H. and A.T., model creation and solution A.R.M. and H.S.G., physiological load calculation A.R.M., G.S. and B.J.F., analysis A.R.M., H.S.G., N.P. and I.R., manuscript preparation A.R.M., N.P., G.S., I.R., V.M., P.H., A.T., B.J.F. and H.S.G., manuscript approval A.R.M., N.P., G.S., I.R., V.M., P.H., A.T., B.J.F. and H.S.G.

## Competing interests

A.R.M., A.T. and H.S.G. are named as inventors on related patent GB2551533, held by jointly University of Bath, Royal Devon & Exeter NHS Foundation Trust and 3D EMS Ltd. The remaining authors have no competing interests.
