## [Peer Review File · Communications Medicine]

Reviewers' comments:

Reviewer #1 (Remarks to the Author):

Manuscript summary:

The paper presents methodology and results from a comparative in silico trial of personalised and generic HTO devices. The trial focused on evaluating and comparing the devices' mechanical safety, specifically by examining peak von Mises stresses developed within the devices and comparing with experimentally determined material fatigue limits. Devices were virtually implanted in 28 patients, subjected to various activity-related loading regimes, and evaluated at three timepoints post-implantation. Aspects of model creation are briefly described, but appear to have been largely validated in a previous study. As the title suggests, the results indicate no difference in safety between the personalised and generic devices.

Overall impression:

The headline result regarding the devices themselves is no doubt interesting in its own right, but the real interest of the paper is the demonstration of the potential of in silico techniques in this domain. I found the work well explained and the results easy to comprehend. The rationale for, and design of the study were clear, including the value in using the in silico approach. The authors have presented a clear template for future in silico studies of this sort.

The trial focused on evaluating mechanical safety, rather than, for example relative efficacy. The latter is probably a more challenging objective, given the wider range of ways in which this could be interpreted/expressed, but is also a question that in silico methods may be better able to address than do conventional trials, especially when long-term efficacy is considered. Similarly, given that in silico methods in principle (computational burden notwithstanding) allow an unlimited variety of device/patient/loading configurations to be examined, one could always criticise the scope of the study (range of activities, range of subjects, etc.). Nonetheless, the study as presented is of great interest as an exemplar of the in silico trials paradigm, of which there remain relatively few.

The study design and results were discussed in detail.

Specific comments/recommendations:

None; the manuscript appears sound to me. A great job!

Reviewer #2 (Remarks to the Author):

The manuscript is clear and complete. Nevertheless it is always nice to have a scientific foundation and this manuscript is such. I have no complaints of the method or the writing as such.

Reviewer #3 (Remarks to the Author):

Thank you so much for giving me a precious opportunity to review this interesting paper. This study evaluated stiffness of personalized HTO plates using a novel in silico virtual assessment. The result showed that there is no increased failure for the new personalized device.

This study is very interesting computational study, but it seems to have some critical flaws from a viewpoint of clinician. What is required of HTO plate is the stability of osteotomy rather than the strength of the plate because most problematic complications are fracture around osteotomy and delayed- and non-union, but not plate breakage. These important complications are related to plate position as well as plate shape and strength. So, surgical procedures and plate design need to be clarified.

Reviewer #4 (Remarks to the Author):

is the patient cohort suitable?

The cohort of patients used in this study seems suitable. When looking to the table S1 we can observe that the average correction angle was 5.99 degrees (from 2.03 to 14.39). This represents common patients who are eligible for HTO. The authors can also provide the frontal HKA angle for each patient which would provide clearer information for the reader. The material properties of each tissue has also been respected.

Are the CT imaging methods appropriate?

The use of a CT scan in the planning of an HTO represents today a more precise and indispensable tool for the realization of an HTO with a custom-made cutting guide. This method is therefore appropriate and necessary. Its non-use would have been a major bias

Are the proposed HTO techniques sound?

As mentioned by the authors in the introduction part, Medial open-wedge HTO for is the most commonly used technique because of its many advantages (complications, consolidation ...) compared to the lateral closing wedge HTO.

The use of the TOMOFIX plate for the generic group is also appropriate because of the very important use of this plate in current clinical practice

Does the research address a genuine problem in the clinic (e.g. mechanical failure of HTO plates), or would the findings inform clinical practice?

One of the main problems with HTO is the loss of the initial correction due to mechanical failure of the osteosynthesis material.

This study is very interesting because it provides biomechanical security arguments on the resistance of personalized plates during daily life activities compared to one of the gold standard: the tomofix plate.

Is the manuscript well written and easy to understand as a clinician?

Yes, very clear and well written. no comment on this point.

Responses to comments in italics

Referee expertise:

Referee #1: Computational biomechanics, in silico trials, finite element analysis

Referee #2: Orthopaedics, HTO

Referee #3: Orthopaedics, HTO

Referee #4: Orthopaedics, HTO, finite element studies

Reviewers' comments:

Reviewer #1 (Remarks to the Author):

Manuscript summary:

The paper presents methodology and results from a comparative in silico trial of personalised and generic HTO devices. The trial focused on evaluating and comparing the devices' mechanical safety, specifically by examining peak von Mises stresses developed within the devices and comparing with experimentally determined material fatigue limits. Devices were virtually implanted in 28 patients, subjected to various activity-related loading regimes, and evaluated at three timepoints post-implantation. Aspects of model creation are briefly described, but appear to have been largely validated in a previous study. As the title suggests, the results indicate no difference in safety between the personalised and generic devices.

Overall impression:

The headline result regarding the devices themselves is no doubt interesting in its own right, but the real interest of the paper is the demonstration of the potential of in silico techniques in this domain. I found the work well explained and the results easy to comprehend. The rationale for, and design of the study were clear, including the value in using the in silico approach. The authors have presented a clear template for future in silico studies of this sort.

The trial focused on evaluating mechanical safety, rather than, for example relative

efficacy. The latter is probably a more challenging objective, given the wider range of ways in which this could be interpreted/expressed, but is also a question that in silico methods may be better able to address than do conventional trials, especially when long-term efficacy is considered. Similarly, given that in silico methods in principle (computational burden notwithstanding) allow an unlimited variety of device/patient/loading configurations to be examined, one could always criticise the scope of the study (range of activities, range of subjects, etc.). Nonetheless, the study as presented is of great interest as an exemplar of the in silico trials paradigm, of which there remain relatively few.

Response:

We thank the reviewer for pointing these issues out and we have modified the Discussion section to explicitly list these limitations. The additional text is given below:

In this in silico trial only mechanical safety and stability were evaluated. It would be interesting to examine relative efficacy as well, which is a question that in silico clinical trials might be able to address better than can conventional clinical trials. However, how to define relative efficacy quantitatively remains an open question – one that would be worth investigating in a future study.

In the current study only a limited number of loading conditions were evaluated. Ideally, a broader range of activities and subjects would have been studied. However, the approach still provides results comparable to those of conventional clinical trial, which would have the same limitations. Furthermore, now that a paradigm for in silico clinical trials is available, a future study could explore mechanical safety and/or relative efficacy for a broader range of activities and subjects.

The study design and results were discussed in detail.

Specific comments/recommendations:

None; the manuscript appears sound to me. A great job!

Reviewer #2 (Remarks to the Author):

The manuscript is clear and complete. Nevertheless it is always nice to have a scientific foundation and this manuscript is such. I have no complaints of the method or the writing as such.

No response required.

Reviewer #3 (Remarks to the Author):

Thank you so much for giving me a precious opportunity to review this interesting paper. This study evaluated stiffness of personalized HTO plates using a novel in silico virtual assessment. The result showed that there is no increased failure for the new personalized device.

This study is very interesting computational study, but it seems to have some critical flaws from a viewpoint of clinician. What is required of HTO plate is the stability of osteotomy rather than the strength of the plate because most problematic complications are fracture around osteotomy and delayed- and non-union, but not plate breakage. These important complications are related to plate position as well as plate shape and strength. So, surgical procedures and plate design need to be clarified.

Response:

We thank the reviewer for their insightful comments. We completely agree that the stability of the osteotomy is an important factor relating to the risk of complications. We did address the issue of HTO plate stability, which contributes to delayed- and non-union problems, by investigating interfragmentary movement (IFM). IFM is a key parameter for osteotomy healing and the prevention of non-union and delayed union [1]. We have modified the Methods and Discussion sections to explicitly state that IFM was used to assess stability. We found that the custom device produced similar levels of IFM to the comparator device for the early healing stages. For healing stage 4 (12 weeks post-operation) the personalised device had 36 microns greater IFM, which is considered to be beneficial.

The surgical technique was identical for the generic and personalised devices, and was virtually performed under the guidance of one of the orthopaedic surgeon co-authors.

With regard to the plate design and position, the key device dimensional parameters i.e width, height, depth, screw hole size, edge radii, and the total number of screws as well as the proximal / distal distribution were consistent for every case. We have clarified the text to highlight both the generic and personalised plate shapes, which are given in Figure S2.

[1] Röderer G, Gebhard F, Duerselen L, Ignatius A, Claes L. Delayed bone healing following high tibial osteotomy related to increased implant stiffness in locked plating. *Injury* 2014;45:1648–52. <https://doi.org/10.1016/j.injury.2014.04.018>.

Reviewer #4 (Remarks to the Author):

is the patient cohort suitable?

The cohort of patients used in this study seems suitable. When looking to the table S1 we can observe that the average correction angle was 5.99 degrees (from 2.03 to 14.39). This represents common patients who are eligible for HTO. The authors can also provide the frontal HKA angle for each patient which would provide clearer information for the reader. The material properties of each tissue has also been respected.

Are the CT imaging methods appropriate?

The use of a CT scan in the planning of an HTO represents today a more precise and indispensable tool for the realization of an HTO with a custom-made cutting guide. This method is therefore appropriate and necessary. Its non-use would have been a major bias

Are the proposed HTO techniques sound?

As mentioned by the authors in the introduction part, Medial open-wedge HTO for is the most commonly used technique because of its many advantages (complications, consolidation ...) compared to the lateral closing wedge HTO. The use of the TOMOFIX plate for the generic group is also appropriate because of the very important use of this plate in current clinical practice

Does the research address a genuine problem in the clinic (e.g. mechanical failure of HTO plates), or would the findings inform clinical practice?

One of the main problems with HTO is the loss of the initial correction due to mechanical failure of the osteosynthesis material.

This study is very interesting because it provides biomechanical security arguments on the resistance of personalized plates during daily life activities compared to one of the gold standard: the tomofix plate.

Is the manuscript well written and easy to understand as a clinician?

Yes, very clear and well written. no comment on this point.

No response required.

REVIEWERS' COMMENTS:

Reviewer #1 (Remarks to the Author):

I've checked through the various materials shared by the authors. I think this constitutes a pretty detailed dataset now. The model from the 2018 paper will show readers precisely how the simulations were instantiated, and the previously submitted files for creating loading configurations, etc., show equally clearly how this model was adapted across the present cohort. The precise study/results reported in the manuscript could of course never be 'exactly' replicated without all of the individual subject models, just as a conventional clinical trial can never be replicated 'exactly' without the same subjects. But, there is ample information here now for researchers to execute a methodologically identical study on a different virtual cohort – for example with differing demographics, or larger cohort – and this seems to me the main point. This will be a valuable resource for researchers in this area.

Reviewer #3 (Remarks to the Author):

no comments

Responses to comments in italics

Reviewers' comments:

Reviewer #1 (Remarks to the Author):

I've checked through the various materials shared by the authors. I think this constitutes a pretty detailed dataset now. The model from the 2018 paper will show readers precisely how the simulations were instantiated, and the previously submitted files for creating loading configurations, etc., show equally clearly how this model was adapted across the present cohort. The precise study/results reported in the manuscript could of course never be 'exactly' replicated without all of the individual subject models, just as a conventional clinical trial can never be replicated 'exactly' without the same subjects. But, there is ample information here now for researchers to execute a methodologically identical study on a different virtual cohort – for example with differing demographics, or larger cohort – and this seems to me the main point. This will be a valuable resource for researchers in this area.

Response:

We thank the reviewer for their comments, and we agree whilst the precise results will not be replicated just as with a conventional clinical trial that the data set and models provided will be helpful for researchers in this area.